# The Effect of *BSCL2* Gene on Fat Deposition Traits in Pigs

**DOI:** 10.3390/ani13040641

**Published:** 2023-02-12

**Authors:** Katarzyna Piórkowska, Julia Sroka, Kacper Żukowski, Karolina Zygmunt, Katarzyna Ropka-Molik, Mirosław Tyra

**Affiliations:** 1Department of Animal Molecular Biology, National Research Institute of Animal Production, Krakowska 1, 32-083 Balice, Poland; 2Department of Biotechnology and Horticulture, University of Agricultural in Kraków, 29-go Listopada 54, 31-425 Kraków, Poland; 3Department of Cattle Breeding, National Research Institute of Animal Production, Krakowska 1, 32-083 Balice, Poland; 4Department of Pig Breeding, National Research Institute of Animal Production, Krakowska 1, 32-083 Balice, Poland

**Keywords:** seipin, *BSCL2*, pig, fat deposition, selective markers, variant calling

## Abstract

**Simple Summary:**

The study evaluated the mutation effects of *BSCL2* variants on slaughter and fattening characteristics and meat quality traits. These mutations were selected based on variant calling analysis and χ test results within subcutaneous fat RNA-seq data. Potential genetic markers revealed significant genotype/allele distribution variations between high- and low-fat pigs. We suggested that the proposed variant calling derived from RNA-seq data might help develop genetic markers for complex pig traits. The results pinpoint that the selection of the *BSCL2* G allele (rs341493267) could increase backfat thickness in pigs. In addition, observed polymorphisms also affected meat percentage, loin mass, and eye area.

**Abstract:**

*BSCL2* encodes seipin, a transmembrane endoplasmic reticulum protein associated with lipodystrophy and severe metabolic complications, including diabetes and hepatic steatosis. In pigs, *BSCL2* expression increases during adipocyte differentiation. In the present study, we identified significant gene variants associated with fat deposition (FD)-related processes based on subcutaneous fat tissue RNA-seq data. In the association study, to prove our hypothesis, three Polish pig breeds were included: Złotnicka White (ZW, *n* = 72), Polish Landrace (PL, *n* = 201), and Polish Large White (PLW, *n* = 169). Based on variant calling analysis and χ^2^ tests, *BSCL2* mutations showing significantly different genotype/allele distribution between high- and low-fat pigs were selected for a comprehensive association study. Four interesting *BSCL2* variants (rs346079334, rs341493267, rs330154033, and rs81333153) belonging to downstream and missense mutations were investigated. Our study showed a significant decrease in minor allele frequency for two *BSCL2* variants (rs346079334 and rs341493267) in PL pigs in 2020–2021. In ZW, *BSCL2* mutations significantly affected loin and ham fats, meat redness, and growth performance traits, such as feed conversion and daily feed intake. Similar observations were noted for PLW and PL, where *BSCL2* mutations influenced fat depositions and meat traits, such as loin eye area, loin mass and fat, carcass yield, and growth performance traits. Based on the observation in pigs, our study supports the theory that *BSCL2* expressed in subcutaneous fat is involved in the FD process.

## 1. Introduction

*BSCL2* is the gene that encodes seipin, a transmembrane endoplasmic reticulum protein. Mutations in the *BSCL2* gene cause congenital lipodystrophy, a rare recessive disorder that appears as a minimum adipose-tissue-level/fat deposition and severe insulin resistance [1]. The gene was discovered in 2001, and various cellular studies were carried out to elucidate its biological function [2]. Seipin is expressed in motor neurons of the spinal cord and cortical neurons of the frontal lobes, as well as adipose tissue (AT). It was shown that seipin deficiency alters lipid droplet morphology and influences adipocyte differentiation. However, the role of this protein remains unclear because the pathophysiology of *BSCL2* in patients with *BSCL2* function disorders is little studied [3]. Nevertheless, it was identified that mutations in *BSCL2* are associated with distal hereditary motor neuropathy (dHMN) and with Charcot–Marie–Tooth disease type 2 (*CMT2*) in a Taiwanese cohort [4]. It was demonstrated that one of these mutations results in low seipin expression and decreased cell viability. Furthermore, in a family of Pakistani heritage, a mutation in *BSCL2* caused lipodystrophy [5]. Individuals affected by mutations in *BSCL2* develop severe metabolic complications, including hepatic steatosis and diabetes. Several mouse studies have shown that Bscl2-deficient mice almost entirely reproduced the lipodystrophy (CGL) phenotype. AT-specific loss due to Bscl2 deficiency was also sufficient to induce generalized early-onset lipodystrophy [6]. *BSCL2* studies have also been conducted on farm animals, including pigs. Kociucka et al. [7] investigated genes (*SNAP23*, *BSCL2,* and *COPA*) involved in lipid droplet formation during porcine adipogenesis in vitro. They observed increased *BSCL2* expression during differentiation and high bscl2 protein level in cells with accumulated lipids.

On the other hand, from a biomedical perspective, pigs are considered an important animal model, particularly for human obesity [8]. In studies predicting fat deposition events, these farm animals reflect human metabolism and fat tissue distribution better than small lab animals. Moreover, pigs are highly economically significant in the meat industry, and AT accumulation is one of the most essential studied processes. AT body expansion/accumulation is the result of two phenomena: hyperplasia, which is the production of new adipocytes, and hypertrophy, which is an increase in the size of existing adipocytes [9]. The excess of circulating fatty acids is converted to triglycerides and stored in adipocyte lipid droplets [10]. In previous years, the importance of the mechanism regulating the expression and formation of lipid droplets has been repeatedly highlighted due to the increasing prevalence of metabolic diseases in the human population, including the spread of obesity. To date, numerous studies searching the genetic background of fatness/obesity have identified many genes involved in significant cellular/molecular processes and pathways [11,12]. However, knowledge in this area still needs to be expanded.

RNA-seq analysis based on next-generation sequencing technology delivers valuable information about transcript levels, including non-coding RNA molecules such as long non-coding RNAs and miRNAs [13]. RNA-seq data can also be a source for identifying gene mutations for fat-storage-related molecular processes [14]. However, a variant calling test based on RNA-seq data requires a much more precise approach, including specialist filtration [15], than a Genome-Wide Association Study (GWAS). Nevertheless, gene variant identification based on RNA-seq data focuses only on genes expressed in this tissue of interest, which could be crucial. Moreover, RNA-seq data in a variant calling analysis can be called an added value because RNA sequencing usually has a different purpose.

Therefore, the present study aimed to identify significant gene variants associated with fat deposition (FD)-related processes based on subcutaneous fat tissue RNA-seq data and to perform an association study to indicate potential genetic markers in the *BSCL2* gene.

## 2. Materials and Methods

The biological material used in the present study was collected after pig slaughtering. Pig slaughtering was performed in the Pig Test Station (PTS, Chorzelów, Poland) of the National Research Institute of Animal Production (NRIAP). The carcasses, after detailed dissection, were intended for sale and consumption. The collection method was non-invasive, so ethical review and approval were unnecessary for this study. However, Approving Experiment Committee of NRIAP (Krakow, Poland) approved all conducted research according to the Polish Act on the Protection of Animals Used for Scientific or Educational Purposes (15 January 2015), which implemented Directive 2010/63/EU of the European Parliament and Council (22 September 2010) on the protection of animals used for scientific purposes. All procedures/methods used in the present study followed the regulations and guidelines of the Local Ethics Committee for Experiments with Animals (Krakow).

### 2.1. Animals and Biological Materials

All Złotnicka White (ZW) female pigs (*n* = 72) were transported to the PTS at NRIAP with a 30 kg initial weight. Following local procedures, their diet and housing conditions were identical [16]. Pigs were fed ad libitum to a 100 ± 2.5 kg final weight. Since 2020, PTS procedures have been slightly changed, and since this year, all pigs are maintained until they reach 120 ± 2.5 kg final weight. This new treatment has been dictated by market demand. The remaining points of the procedure continue to follow the pre-2020 guidelines. Detailed growth information, such as feed conversion, daily gain, number of test days, and feed intake, was collected during a test. Before slaughter, pigs were starved for 24 h. After slaughter, carcasses were chilled for 24 h at 4 °C, and then, the right half-carcass was evaluated. During the dissection, several carcass characteristics were measured, including loin (kg) and ham mass (kg), average backfat thickness (cm) (from five measurements—backfat thickness over the shoulder blade, at the thoracic part, and three measures at the sacrum part of the spine), loin eye area (cm^2^) (between the last thoracic and first lumbar vertebras), carcass yield (%), and numerous other fat tissue related traits, including visceral and subcutaneous fat content (peritoneal fat, ham, knuckle and loin fat with skin) [16]. Meat quality characteristics (intramuscular fat, pH, meat color, and myowater exudation) were assessed according to Tyra and Żak protocol [17].

From all 72 ZW pigs, subcutaneous fat tissue and blood samples were collected up to 20 min after slaughter. The fat samples were stabilized in RNAlater™ Stabilization Solution (Applied Biosystems, Waltham, MA, USA), and the blood was stabilized in EDTA tubes; all was frozen at −20 °C. The pigs for the transcriptomic study were selected based on fat-related trait information collected during detailed dissection. We created two pig groups, showing low- (*n* = 8, LFD) and high-fat (*n* = 8 HFD) deposition levels, following the approach of Piórkowska et al. [18].

The association study included female ZW (*n* = 72), Polish Landrace (PL) (*n* = 201), and Polish Large White (PLW) (*n* = 169) pigs. The material for PLW and PL pigs was collected between 2020 and 2021 to show selective potential in the present populations. ZW is a native Polish breed, and PL and PLW belong to maternal components in Polish breeding. DNA was isolated from tissue and blood samples depending on the available biological material. Blood samples from PLW and PL pigs were collected by the Department of Pig Breeding of NRIAP as part of other research projects.

### 2.2. RNA Sequencing and Transcript Variant Identification

RNA isolation from fat tissue, next-generation sequencing, and raw data processing were described previously by Piórkowska et al. [18]. Alignment of raw reads to the pig transcriptome was performed with the pig reference genome Sscrofa11.1 (GCA_000003025.6, Ensembl 106: Apr 2022). In our previous study [19], where variant calling used liver RNA-seq data to identify transcript variants, we used GATK v. 4.1.9 and Picard [20] tools. In the present study, a similar approach was applied. The current RNA-seq data was deposited in Gene Expression Omnibus NCBI (GSE160436). The functional gene annotation clustering and protein–protein interaction (PPI) analyses (for genes with identified missense variants) were performed using DAVID Functional Annotation Bioinformatics Microarray Analysis [21] and String database [22], respectively. Moreover, we estimated the gene expression pattern for each *BSCL2* isoform (*ENSSSCT00000025488, ENSSSCT00000057466, ENSSSCT00000055749, ENSSSCT00000061218*) (Ensembl release 106: Apr 2022) based on Fragments Per Kilobase of transcript per Million mapped reads (FPKM) generated in RNA-seq analyses.

Four interesting *BSCL2* variants (rs346079334, rs341493267, rs330154033, and rs81333153) belonging to downstream (ds) and missense mutations (ms) were selected for association study because they presented the most significant differences in allele/genotype distribution according to the χ^2^ test. Moreover, identified *BSCL2* ms variants were analyzed by the GERP [23] and SIFT tools [24], which defined the detriment of these mutations. Furthermore, these results were validated by DNA sequencing using the Sanger method. All *BSCL2* primers are presented in Appendix A.

### 2.3. BSCL2 Genotyping, Frequency Calculation and Statistical Analyses

PCR-RFLP (*AluI* restriction enzyme) and Sanger sequencing were used for genotyping *BSCL2* downstream (rs346079334) and three missense (rs341493267, rs330154033, and rs81333153) mutations, respectively. The association study included ZW (*n* = 72), PL (*n* = 201), and PLW (*n* = 169) pigs. Examining *BSCL2* variant frequencies allowed us to determine the selection potential targeted to increase the pork quality lost in the previous intense breeding.

The first test included 16 ZW pigs and used the ANOVA procedure (SAS v. 7.1 with default settings; SAS Institute, Cary, NC, USA) with a post-hoc Duncan test to indicate differences in pig production traits dependent on *BSCL2* genotypes.

Next, comprehensive association analysis including 72 ZW, 201 PL, and 169 PLW pigs was conducted based on the GLM procedure (SAS v. 8.02 with default settings; SAS Institute, Cary, NC, USA). All pigs involved in analyses were free of a deleterious mutation in the *RYR1* gene [25].

In this analysis, we used the presented linear model:Y_ijkl_ = μ+d_i_ + c_j_ + f_j_ + α_(xijk )_ + e_ijkl_

The model’s terms are as follows: Y_ijkl_—observation;µ—mean;d_i_—fixed effect of genotype group;c_j_—fixed effect of sire;f_j_—fixed effect of farm origin;α_(xijk)_—covariate for the weight of the right side of the carcass;e_ijkl_—random error.

The model did not include the effect of the slaughter season because temperature, humidity, and feeding conditions were the same for all pigs, independent of the season. The means were presented as least-square means (LSM ± SE) to determine significant differences between genotype groups. Each breed was analyzed individually. Moreover, additive and dominance effects were calculated (regression procedure, SAS v. 8.02, default setting, SAS Institute, Cary, NC, USA). The additive effect oscillated between −1 and 1 for AA and GG genotypes, and the dominance impact oscillated between −1 and 1 for AG and both homozygotes.

## 3. Results

### 3.1. Animals Characteristics and Variant Calling Results

Animal groups (HFD and LFD) used in RNA-seq analysis were significantly different in the context of fat deposition traits, which was previously described [18].

The information about read numbers after NGS, mapping, annotation, and also clustering into low- and high-fat groups was presented in a previous study [18]. GATK analysis showed that based on fat RNA-seq data, over 60,000 mutations were identified, and 5870 were found to be significant according to the χ^2^ test, which means that these differences in genotype/allele distribution between HFD and LFD were crucial. Over 70% of identified variants were known (deposited in the dbSNP NCBI and Ensembl databases). Insertions and deletions constituted 8.4%, of which 289 were significant following the χ^2^ test. In total, 323 and 2500 significant gene mutations were found in the 5′UTRs and 3′UTRs, respectively. Off-gene variants (down- and upstream and intergenic) were numerous: over 40,000 (Table 1).

### 3.2. Genes with Missense Variants Functional Analysis

The functional analysis performed by DAVID indicated that genes with missense mutations are involved in the regulation of inflammatory response (GO:0050727)(*AKNA, TNIP1, SETD6, DROSHA, ADAMTS12*), fat cell differentiation (GO:0045444)(*BBS1, TMEM120B, MED1, MKKS*), cellular response to insulin stimulus (GO:0032869)(*DENND4C, SLC2A4, COMMD6, LPIN1*), calcium ion binding (GO:0005509)(*CEMIP2, CLSTN3, VWCE, PKD2, ZZEF1, VCAN, EGFLAM, SGCA, S100A16, CDHR4, SPARCL1, DCHS1, MAN1A1, CAPN1, PAMR1, NINL, NUCB2, SLC25A24, FBN1),* and cell adhesion molecules *(ssc04514)(SLA-DRB1, VCAN, ITGAM, CD6, SLA-7, SLA-3*) (Figure 1A). Moreover, the top 15 genes based on a chi^2^ test, which contained missense variants, were associated with lipid metabolic processes, fat cell structure, and adipocyte differentiation (*BSCL2, DHRS11, TMEM120B*) (Figure 1B). For further analysis, the *BSCL2* mutations were chosen. PPI STRING analysis confirmed a significant *BSCL2* role and its involvement in lipid droplet formation (Figure 2A). Moreover, based on RNA-seq data, we compared the expression of *BSCL2* isoforms dependent on FD, and we found significant differences and trends (Figure 2C). Thus, we concluded that these isoforms could play slightly distinct molecular roles.

### 3.3. Influence of BSCL2 Mutations on Pig Phenotypes

We identified 42 *BSCL2* variants, including missense and synonymous mutations in the 3′UTR, 5′UTR, upstream, downstream, intergenic, and intron regions. Nevertheless, only 15 were significant, according to the χ test. All three ms (rs341493267, rs330154033, rs81333153) variants and one ds (rs346079334) variant were selected for further association analysis; they showed 0.00055, 0.037581251, 0.192933798, and 0.00055 *p*-values, respectively. The presence of these variants was also confirmed by Sanger sequencing (Figure 2B). Moreover, GERP pinpointed a low score of −6.34 for the rs330154033 mutation, and the SIFT defined it as deleterious.

In ANOVA analysis of 16 ZW pigs, it was shown that fully coupled rs341493267 and rs346079334 *BSCL2* variants significantly affected body composition traits, mainly associated with fat storage, such as subcutaneous fats measured in numerous carcass points and also peritoneal fat (*p*-value = 0.001) (Table 2). In addition, missense variants rs330154033 and rs81333153 were also related to body traits, including meat percentage, carcass yield, and mass of primary cuts (Table 3 and Table 4).

### 3.4. BSCL2 Frequency in Polish Pig Populations and Comprehensive Association Analysis

After genotyping, there were high differences in *BSCL2* variant frequency between ZW, PL, and PLW pigs. The C allele of rs81333153 mutation was dominant in Złotnicka White, and in maternal breeds, the G allele prevailed (Appendix A). Moreover, during two years (2020–2021) observation, the number of pigs with TT (rs346079334) and AA (rs341493267) genotypes in Polish Landrace decreased twofold (14–7%) and fourfold (8–2%) (Appendix A), respectively. In turn, the number of pigs with CC genotype (rs81333153) increased both in PL and PLW.

Analysis of *BSCL2* haplotypes showed that the two most numerous haplotypes in ZW pigs were GG/GG/GG/CC and GT/AG/AG/CG (31%); in PL, the most numerous haplotype was GG/GG/GG/GG (21%), and in PLW pigs, the most frequent haplotype was under 15%.

The association study in Złotnicka White pigs demonstrates that the ds variant (rs346079334) significantly affected loin and ham fats, with TT pigs showing 10 and 4% higher values than CC, respectively. Moreover, a significant influence on meat redness was observed, where heterozygote meat was the reddest. The three missense variants of the *BSCL2* gene also significantly affected loin and ham fat depositions; rs330154033 and rs81333153 mutations also changed growth traits, such as feed conversion and daily feed intake, respectively (Table 5). In the Appendix A, we also attach tables presenting trends that suggest that *BSCL2* influences fat deposition in Złotnicka White (Appendix A). Association analysis, including two maternal breeds PL and PLW, demonstrated numerous dependencies between *BSCL2* variants and fat deposition, though statistical analysis in these breeds was complicated due to the low number of alternate homozygotes. In these breeds used in Polish breeding, *BSCL2* variants, in addition to affecting fat characteristics, also influenced meat traits such as mass and eye area of loin and carcass yield (Table 6 and Table 7). In addition, in PLW, numerous relationships between *BSCL2* and growth performance traits (daily gain, feed conversion, and slaughter age) were observed (rs341493267–AA, rs330154033–AA, and rs81333153–CC, respectively). Appendix A presenting trends for PLW and PL pigs also suggest that *BSCL2* regulates fat deposition processes (Appendix A). In PL pigs, TT_rs346079334 and AA_rs341493267 genotypes correlated with a higher subcutaneous fat level; nevertheless, decreasing the number of pigs with these genotypes suggests the still-unfavorable direction of the conducted selection.

## 4. Discussion

### 4.1. Adipogenesis in Terms of the Role of the BSCL2 Gene in Fat Deposition

Adipose tissue (AT) is crucial to the body’s energy and metabolic processes because it stores fatty acids. This tissue is involved in the effective functioning of the endocrine and immune systems (producing or activating hormones) and in tissue regeneration processes due to the presence of progenitor cells with multipotent differentiation potential [26]. AT can be divided into two types: white adipose tissue (WAT), which plays a role in metabolic and energy processes, as it contains large and single localized lipid droplets, and brown adipose tissue (BAT), which is involved in thermoregulation of the body and is morphologically rich in mitochondria, in which uncoupling protein-1 (UCP-1) is localized [27]. The expansion of AT due to hyperplasia is the basis of adipogenesis, a process actively controlled by many genes and signaling pathways. Adipocytes make up to 40% of cells in the WAT [28]. Adipogenesis occurs in two stages: in the first, mesodermal stem cells (MSCs) with multipotent differentiation potential are transformed into pre-adipocytes [29]. In the subsequent phase of late differentiation, the pre-adipocytes differentiate into mature adipocytes upon activation of the PPAR-gamma (Peroxisome proliferator-activated receptor gamma) receptor. The maintenance of expression of adipocyte marker genes is controlled by PPAR-gamma interacting with C/EBP (CCAAT/enhancer-binding protein) [30]. However, adipogenesis involves many more genes and signaling pathways—such as Wnt, Hedgehog, c-AMP (cyclic adenosine monophosphate), IGF-1 (insulin growth factor 1), and BMP-2, -4 (bone morphogenetic protein 2, 4) [31].

In the present study, mutations in the *BSCL2* gene were significantly associated with fat deposition in ZW pigs, which was suggested based on variant calling analysis using RNA-seq data. In the *BSCL2* gene, 42 single-nucleotide mutations (including 15 that were significant according to χ2 test) were identified across pigs with different fat deposition. This observation could support the hypothesis that *BSCL2* plays a crucial role during adipogenesis. *BSCL2*, through the cAMP/PKA pathway, participates in regulating the processes in WAT and the differentiation and maintenance of adipocytes [32]. Zhou et al. [33] suggest that in vivo *BSCL2* controls, at least to some extent, the adipogenesis process through the enzyme ATGL (adipose triglyceride lipase), which is involved in the initial phase of lipogenesis. ATGL was previously studied in the context of fat management and food intake regulation, and the authors found a significant effect on fattening traits in pigs [34]. The knockout of the *BSCL2* gene in mice in vitro disrupted the late adipogenesis process and expressed an increase in lipid droplet formation. It seems likely that *BSCL2*-knockout mice did not show adipogenesis inhibition but showed lipogenesis suppression [35]. Our analysis using the STRING to predict PPI indicates that the *BSCL2* gene is involved in forming lipid droplets. Moreover, the present study found that the expression of particular *BSCL2* isoforms in SAT varied dependent on fat deposition level. Regarding *BSCL2* mutations investigated in the present study, two missense variants (rs330154033 and rs81333153) are located in the IPR009617 protein domain, which is the seipin family domain, whose primary function is to control the adipogenesis process by regulating lipolysis in a cell-independent manner [32]. Therefore, these mutations can play a crucial role in the proper function of *BSCL2* protein; in addition, one of them (rs330154033) was defined by SIFT and GERP as highly unfavorable or deleterious. Moreover, two others located in the last exon (rs341493267) and downstream of the gene (rs346079334) can play a regulatory role during expression.

*BSCL2* was investigated regarding processes related to obesity/fatness, which is defined as excessive accumulation of triacylglycerol in WAT, specifically in lipid droplets. Our study also contributes to knowledge about the connection between *BSCL2* and the FD process using pigs as an animal model. It supports the hypothesis that porcine adipocytes are excellent and reliable material for studying adipogenesis due to remarkably high similarity to human adipocytes, overtaking small lab animals in this competition [36].

### 4.2. BSCL2 Variants as Potential Selective Markers for Complex Pig Traits

The accumulation of AT considerably impacts the growth efficiency and quality of meat in pig livestock, and understanding the molecular basis of this process is critical to improving production [8]. In previous liver RNA-seq studies, we proved that the *FGL1* (fibrinogen-like 1) gene is associated with FD in pigs, and the rs340465447_A allele may be a selectable marker for fat level [19]. By analogy, this study searched for confirmation that the *BSCL2* gene variants may also play a similar role, and we found numerous relationships in all investigated pig breeds, though statistical analysis in PLW and PL breeds was complicated due to the low number of individuals in alternate homozygote groups. Our results pinpoint that the selection of the *BSCL2* G allele (rs341493267) could increase backfat thickness in pigs.

The porcine *BSCL2* gene is located on chromosome 2 in the p-arm (SSC2p) region, which was identified as a region of QTL distribution for traits such as the diameter of adipocytes [5], which was supported by the report of Kociucka et al. [7]. To date, many fat-related loci in pigs have been mapped to SSC2. In Landrace and Korean native pigs, QTL on SSC2 was correlated with backfat thickness (BFT) [37]. A study in Duroc pigs showed the presence of QTLs for attributes such as total cholesterol content, low-density lipoprotein (LDL), and triacylglycerol content [38]. Returning to *BSCL2*, in a genome-wide detection of copy number variation (CNV) study, several CNVs associated with daily gain overlapped with the *BSCL2* gene, supporting the theory that *BSCL2* is a promising functional candidate gene for complex pig traits [39].

## 5. Conclusions

The present report shows that individual variants of the *BSCL2* gene were related to features of fat and meat in three Polish pig breeds. Our study supports the hypothesis that *BSCL2* is involved in determining fat deposition in pigs and can be used as a selective marker to modulate fat levels in this species. However, our observation in PL pigs concerning the number of TT_rs346079334 and AA_rs341493267 genotypes, which have been correlated with a higher subcutaneous fat level, still suggests the unfavorable direction of the conducted selection, despite the boosted awareness of insufficient fatness in pigs. Moreover, the present study confirms that RNA sequencing could be adequate data for variant calling analysis, combined with comprehensive association analysis, which validates the observations.

## Figures and Tables

**Figure 1 animals-13-00641-f001:**
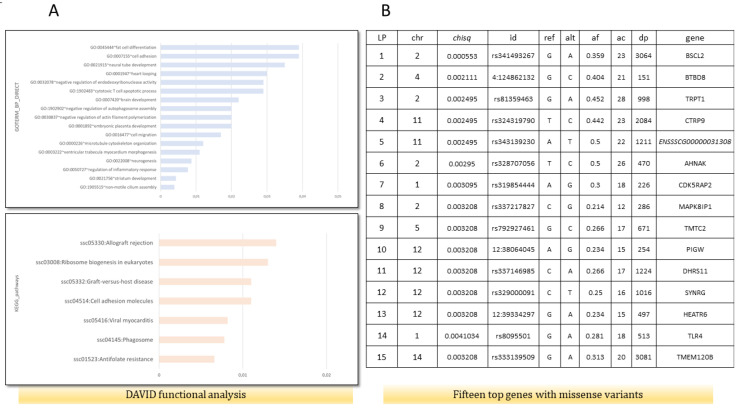
Functional analysis for genes containing missense variants (ms) and genes with the lowest corrected *p*-value in χ^2^ test drawn from variant calling analysis. (**A**) Functional analysis using DAVID, including GO biological process and KEGG pathways on 363 genes with identified missense mutations in fat RNA-seq data. (**B**) Top 15 genes with ms according to χ^2^ test. Chr—chromosomal, id—Ensembl number, reference and alternate alleles, af—alternate allele frequency, dp—depth of sequencing, and gene names.

**Figure 2 animals-13-00641-f002:**
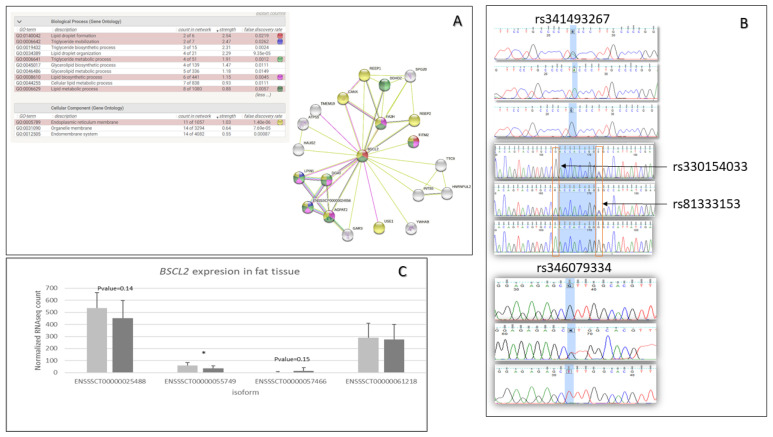
*BSCL2* gene information about the gene function, *BSCL2* isoform expression, and *BSCL2* identified polymorphisms. (**A**) PPI analysis for *BSCL2* gene using STRING tool. (**B**) Chromatograms present Sanger sequencing results for three missense mutations (rs341493267, rs330154033, rs81333153) and one downstream gene variant (rs346079334). The chromatograms were visualized by FinchTV vs. 1.4.0 chromatogram viewer (Geospiza, Inc.; Seattle, WA, USA; http://www.geospiza.com), accessed on 20 April 2022. (**C**) *BSCL2* isoform expression dependent on fat tissue deposition based on fat RNA-seq results. Light grey—LFD group, dark grey—HFD.

**Table 1 animals-13-00641-t001:** Variant calling results generated based on subcutaneous fat transcriptome data.

Mutation (Gene Variant)		Quantity
	Total	χ^2^ Test *
Total	69,102	5870
Known variant % (having “rs” number)	73.2	76.5
Analyzed genes	12,473	2598
Insertion	3220	180
Deletion	2597	109
SNP	63,283	19,789
High effect	1933	43
Low effect	11,675	1070
Moderate effect	4538	370
Missense variant	4400	363
Silent variant	10,570	994
3′UTR	25,550	2497
5′UTR with the premature start codon	576	55
5′UTR	3777	323
Frameshift variant	1134	25
Splice region variant	1386	45
Inframe variant	51	8
Downstream gene	25,849	2000
Upstream gene	10,867	865
Intergenic region	7641	580
Intron	23,612	1733
Start lost	9	2
Stop gain	27	3
Stop lost	14	1

Legend: * significant differences in allele/genotype distribution based on χ^2^ test; the differences were calculated between HFD and LFD groups. Low, moderate, and high effects—the power of effects on protein expression, phenotype, and amino acid sequence. Definitions of high, low, and moderate effects can be found at http://snpeff.sourceforge.net/SnpEff_manual.html (accessed on 20 April 2022).

**Table 2 animals-13-00641-t002:** Means with standard deviation (SD), calculated based on ANOVA test (SAS) for pig production traits dependent on rs341493267 and rs346079334 *BSCL2* variants, for 16 Złotnicka White pigs included in RNA sequencing analysis.

Traits	*BSCL2* Genotype			
	AA (3) TT	AG (7) GT	GG (6) GG	*p*-Value
Daily gain (g)	767 ± 0.71	721 ± 0.99	649 ± 0.30	0.068
All forage used (kg)	234 ± 26 ab	236 ± 30 a	268 ± 20 b	0.03
Carcass yield (kg)	74.70 ± 1.56ab	74.36 ± 1.18 a	76.03 ± 1.50 b	0.036
Loin fat with skin (kg)	1.71 ± 0.13 A	1.94 ± 0.62 ABa	2.55 ± 0.15 Bb	0.0007
Ham fat with skin (kg)	1.97 ± 0.16 A	1.96 ± 0.41 A	2.53 ± 0.23 B	0.008
Peritoneal fat (kg)	0.46 ± 0.05 A	0.57 ± 0.21 ABa	0.77 ± 0.12 Bb	0.001
Backfat over shoulder (cm)	2.10 ± 0.42 a	2.70 ± 0.67 ab	2.95 ± 0.33 b	0.04
Backfat over lumbar I	1.53 ± 0.25 A	2.01 ± 0.72 AB	2.52 ± 0.30B	0.005
Backfat over lumbar II	1.43 ± 0.13 A	1.79 ± 0.74 B	2.12 ± 0.33 B	0.003
Backfat over lumbar III	1.93 ± 0.09 Aa	2.16 ± 0.61 B	2.60 ± 0.65 b	0.03
Backfat thickness (cm)	1.73 ± 0.04 A	2.11 ± 0.64 B	2.47 ± 0.26 B	0.0005
Feet mass (kg)	1.08 ± 0.05 Aa	0.97 ± 0.06 ABb	0.91 ± 0.07 B	0.008
Loin mass without skin and fat (kg)	4.98 ± 0.53	4.68 ± 0.34	4.40 ± 0.26	0.07
Ham mass (kg)	9.28 ± 0.57 ab	8.63 ± 0.57 a	9.2 ± 0.37 b	0.04
Knuckle fat with skin (kg)	0.21 ± 0.01 A	0.23 ± 0.03 AB	0.24 ± 0.02 B	0.007
Loin eye height (cm)	6.7 ± 0.081A	6.5 ± 076 AB	6.22 ± 0.28 B	0.005
Meat percentage %	55.8 ± 2.18 a	52.8 ± 5.01 ab	49.8 ± 2.09 b	0.02
Backfat at the point K1	1.60 ± 0.08 A	1.87 ± 0.76 AB	2.48 ± 0.50 B	0.005
Meat percentage in primary cuts (kg)	64.3 ± 2.21 A	60.7 ± 5.42 AB	57.7 ± 2.63 B	0.01
pH24	5.43 ± 0.02 AB	5.40 ± 0.04 A	5.47 ± 0.04 B	0.009

The same superscripts described values belonging to the same statistical group (A,B = *p* < 0.01; a,b = *p* < 0.05). K1 point—backfat thickness calculated in cm over the lateral edge of longissimus dorsi muscle.

**Table 3 animals-13-00641-t003:** Means with standard deviation (SD), calculated based on ANOVA test (SAS) for pig production traits dependent on rs330154033 *BSCL2* variant, for 16 Złotnicka White pigs included in RNA sequencing analysis.

Traits	*BSCL2* Genotype			
	AA (3)	AG (8)	GG (5)	*p*-Value
Carcass yield (kg)	74.70 ± 1.56 AB	74.2 ± 1.17 A	76.6 ± 0.88 B	0.0016
Peritoneal fat (kg)	0.46 ± 0.05 A	0.57 ± 0.20 ABa	0.81 ± 0.10 Bb	0.0007
Feet mass (kg)	1.09 ± 0.05 Aa	0.96 ± 0.06 ABb	0.90 ± 0.07 B	0.0076
Loin mass (kg)	6.69 ± 0.53 ab	6.64 ± 0.44 a	6.99 ± 0.12 b	0.04
Loin fat with skin (kg)	1.71 ± 0.13 A	2.01 ± 0.61 ABa	2.56 ± 0.17 Bb	0.006
Ham fat with skin (kg)	1.97 ± 0.22 a	2.13 ± 0.47ab	2.54 ± 0.17 b	0.04
Knuckle fat with skin (kg)	0.24 ± 0.02 a	0.24 ± 0.03 a	0.21 ± 0.01 b	0.019
Backfat over lumbar I	1.53 ± 0.25 A	2.06 ± 0.68 AB	2.54 ± 0.32B	0.004
Backfat over lumbar II	1.43 ± 0.13 A	1.81 ± 0.69 AB	2.14 ± 0.35 B	0.007
Backfat thickness (cm)	1.73 ± 0.04 A	2.16 ± 0.61 AB	2.47 ± 0.29 B	0.003
Loin eye height (cm)	6.70 ± 0.081A	6.49 ± 0.72 AB	6.18 ± 0.29 B	0.01
Loin mass without skin and fat (kg)	4.98 ± 0.53	4.68 ± 0.34	4.40 ± 0.26	0.07
Backfat at the point K1	1.60 ± 0.08 A	1.92 ± 0.72 AB	2.52 ± 0.54 B	0.01
Primary cuts (kg)	64.3 ± 2.21 A	60.3 ± 5.17 AB	57.7 ± 2.63 B	0.01
Meat percentage %	55.8 ± 2.18 a	52.8 ± 5.01 ab	49.8 ± 2.09 b	0.02
pH24	5.44 ± 0.02 AB	5.41 ± 0.04 A	5.47 ± 0.04 B	0.02

The same superscripts described values belonging to the same statistical group (A,B = *p* < 0.01; a,b = *p* < 0.05). K1 point—backfat thickness calculated in cm over the lateral edge of longissimus dorsi muscle.

**Table 4 animals-13-00641-t004:** Means with standard deviation (SD), calculated based on ANOVA test (SAS) for pig production traits dependent on rs81333153 *BSCL2* variant, for 16 Złotnicka White pigs included in RNA sequencing analysis.

Traits	*BSCL2* Genotype			
	CC	CG	GG	*p*-Value
Daily gain	638 ± 29 a	710 ± 90 b	761 ± 71 b	0.04
Peritoneal fat (kg)	0.69 ± 0.06 A	0.65 ± 0.23 ABa	0.46 ± 0.05 Bb	0.008
Feet mass (kg)	0.95 ± 0.03 a	0.94 ± 0.08 a	1.09 ± 0.05 b	0.013
Loin mass (kg)	7.13 ± 0.02 A	6.67 ± 0.39 B	6.69 ± 0.53 AB	0.003
Loin fat with skin (kg)	2.41 ± 0.07 A	2.17 ± 0.62 ABa	1.71 ± 0.13 Bb	0.003
Ham fat with skin (kg)	2.31 ± 0.10 a	2.20 ± 0.50 ab	1.97 ± 0.17 b	0.04
Knuckle fat with skin (kg)	0.24 ± 0.02 ab	0.23 ± 0.03 a	0.21 ± 0.01 b	0.018
Backfat over lumbar I	2.37 ± 0.26 a	2.21 ± 0.68 a	1.53 ± 0.25 b	0.016
Backfat over lumbar II	2.10 ± 0.29 a	1.89 ± 0.66 ab	1.43 ± 0.12 b	0.03
Backfat thickness (cm)	2.34 ± 0.14 A	2.26 ± 0.60ABa	1.73 ± 0.04 Bb	0.008
Loin mass (kg)	7.56 ± 0.25a	6.79 ± 0.37 b	7.57 ± 0.40 a	0.01
Backfat at the point K1	2.47 ± 0.33 a	2.06 ± 0.76 ab	1.6 ± 0.08 b	0.03
Meat percentage in primary cuts (kg)	59.5 ± 0.30 a	59.3 ± 5.27 a	64.3 ± 2.21 b	0.03
Meat percentage %	51.6 ± 2.18 ab	51.4 ± 4.80 a	55.8 ± 2.17 b	0.04

**Table 5 animals-13-00641-t005:** Means as least-square means (LSM) with standard error (SE) for pig production traits dependent on *BSCL2* genotypes for Złotnicka White.

Mutation	Traits	Genotype						GLM Significance				Effect	
		LSM	SE	LSM	SE	LSM	SE	*BSCL2* *p*-Value	X2P	Farm	Sire	Additive	Dominance
		AA (*n* = 11)		AG (*n* = 27)		GG (*n* = 34)						A→G	het→hom
*rs341493267*	Feet mass (kg)	0.98 ^a^	0.03	0.94 ^ab^	0.01	0.92 ^b^	0.01	0.0496	**	ns	ns	−0.03 *	-
	Loin fat with skin (kg)	1.89 ^AB^	0.13	1.93 ^A^	0.12	2.13 ^B^	0.09	0.0048	**	*	ns	-	-
	Ham fat with skin (kg)	2.04 ^ab^	0.09	2.03 ^a^	0.08	2.16 ^b^	0.09	0.0125	**	*	ns	-	-
	Loin mass without skin and fat (kg)	4.46 ^ab^	0.17	4.53 ^a^	0.10	4.35 ^b^	0.08	0.0395	***	ns	ns	-	-
	Backfat at lumbar I (cm)	1.97 ^a^	0.21	2.11 ^ab^	0.11	2.34 ^b^	0.09	0.0462	**	*	ns	+0.19 ^T^	-
	Backfat at lumbar II (cm)	1.83 ^ab^	0.19	1.86 ^a^	0.11	2.05 ^b^	0.09	0.0413	**	*	ns	-	-
*rs346079334*		GG (*n* = 11)		GT (*n* = 24)		TT (*n* = 37)							
	Feet mass (kg)	0.98 ^a^	0.03	0.95 ^ab^	0.01	0.92 ^b^	0.01	0.0427	***	ns	ns	−0.03 *	-
	Loin fat with skin (kg)	1.89 ^ab^	0.13	2.00 ^a^	0.13	2.07 ^b^	0.09	0.0167	*	*	ns	-	-
	Ham fat with skin (kg)	2.04 ^a^	0.09	2.08 ^ab^	0.08	2.12 ^b^	0.08	0.0493				-	
	Meat color redness	15.8 ^ab^	0.21	15.1 ^a^	0.19	15.6 ^b^	0.29	0.0492	-	-	ns	-	+0.30 ^T^
*rs330154033*		AA (*n* = 11)		AG (*n* = 28)		GG (*n* = 33)							
	Feet mass (kg)	0.98 ^a^	0.03	0.94 ^ab^	0.11	0.92 ^b^	0.01	0.0488	***	*	ns	+0.03 *	-
	Loin fat with skin (kg)	1.89 ^a^	0.13	1.97 ^ab^	0.12	2.11 ^b^	0.10	0.0125	**	*	ns	-	-
	Ham fat with skin (kg)	2.04 ^a^	0.09	2.06 ^ab^	0.08	2.14 ^b^	0.09	0.0387					
	Feed conversion (kg/kg)	3.21 ^a^	0.11	3.39 ^ab^	0.10	3.45 ^b^	0.09	0.0477				-	+0.12 ^T^
*rs81333153*		GG (*n* = 12)		GC (*n* = 37)		CC (*n* = 23)							
	Loin mass (kg)	6.57 ^a^	0.12	6.40 ^b^	0.09	6.40 ^b^	0.19	0.0085	***	ns	ns	-	-
	Loin fat with skin (kg)	1.93 ^a^	0.12	2.03 ^ab^	0.10	2.07 ^b^	0.11	0.0224	***	**	ns	-	-
	Backfat at lumbar II (cm)	1.82 ^a^	0.17	1.94 ^b^	0.09	2.03 ^ab^	0.13	0.0468	**	*	ns	-	-
	Daily feed intake (kg)	2.37 ^ab^	0.04	2.42 ^b^	0.04	2.24 ^a^	0.06	0.0359	-	-	*	-	+0.14 ^T^

Mean and standard error were estimated using general linear model. The same superscripts described values belonging to the same statistical group (A,B = *p* < 0.01; a,b = *p* < 0.05). *p*-value in GLM significant * *p* < 0.05, ** *p* < 0.01, *** *p* < 0.001, ns—not significant. K1 point—backfat thickness calculated in cm over the lateral edge of longissimus dorsi muscle. T—trend, *p*-value 0.05–0.20, het—heterozygous, hom—homozygous.

**Table 6 animals-13-00641-t006:** Means as least-square means (LSM) with standard error (SE) for pig production traits dependent on BSCL2 genotypes for Polish Landrace.

Mutation	Traits	Genotype						GLM Significance				Effect	
		LSM	SE	LSM	SE	LSM	SE	*BSCL2* *p*-Value	X2P	Farm	sire	Additive	Dominance
		AA (*n* = 9)		AG (*n* = 86)		GG (*n* = 105)						A→G	het→hom
*rs341493267*	Knuckle fat with skin (kg)	1.30 ^a^	0.02	1.37 ^ab^	0.01	1.39 ^b^	0.01	0.0437	**	*	ns	-	+0.044 *
	Backfat at the point K1	1.70 ^AB^	0.50	1.63 ^A^	0.08	1.35 ^B^	0.05	0.0447	***	ns	ns	-	−0.2 3*
	Feed conversion (kg/kg)	2.93 ^ab^	0.13	2.90 ^a^	0.04	2.94 ^b^	0.05	0.0471	-	ns	*	−0.22 *	+0.13 *
	Slaughter age (days)	190 ^a^	8	188 ^b^	2	188 ^b^	1	0.0490	-	ns	ns	-	-
	Days in test (days)	107 ^a^	5	103 ^b^	1	104 ^b^	1	0.0224	-	ns	ns	-	-
	Loin pH24	5.49 ^a^	0.02	5.54 ^b^	0.01	5.52 ^b^	0.008	0.0153	-	ns	ns	+0.044 *	−0.016 ^T^
*rs346079334*		GG (*n* = 108)		GT (*n* = 73)		TT (*n* = 20)							
	Feed conversion	2.92 ^ab^	0.04	2.94 ^a^	0.04	2.87 ^b^	0.06	0.0471	-	ns	*	-	-
	Loin pH24	5.48 ^a^	0.02	5.5 ^ab^	0.01	5.50 ^b^	0.009	0.0153	-	ns	ns	+0.04 *	−0.028 **
*rs330154033*		AA (*n* = 12)		AG (*n* = 68)		GG (*n* = 120)							
	Knuckle fat with skin (kg)	1.37 ^ab^	0.05	1.34 ^a^	0.01	1.40 ^b^	0.01	0.0161	*	**	ns	-	+0.02 ^T^
	Backfat at the point C1	1.90 ^a^	0.37	1.49 ^ab^	0.06	1.46 ^b^	0.05	0.0292	***	**	ns	−0.23 *	+0.095 ^T^

Mean and standard error were estimated using a general linear model (GLM). The same superscripts described values belonging to the same statistical group (A,B = *p* < 0.01; a,b = *p* < 0.05). *p*-value in GLM significant * *p* < 0.05, ** *p* < 0.01, *** *p* < 0.001, ns—not significant. K1 point—backfat thickness calculated in cm over the lateral edge of longissimus dorsi muscle. C1—backfat thickness calculated in cm in height extension of loin eye. T—trend, *p*-value 0.05–0.20, het—heterozygous, hom—homozygous. Loin pH24 – pH in loin measured 24 h after slaughter.

**Table 7 animals-13-00641-t007:** Means as least-square means (LSM) with standard error (SE) for pig production traits dependent on BSCL2 genotypes for Polish Large White.

Mutation	Traits	Genotype						GLM Significance				Effect	
		LSM	SE	LSM	SE	LSM	SE	*BSCL2*	X2P	Farm	Sire	Additive	Dominance
		AA (*n* = 8)		AG (*n* = 70)		GG (*n* = 91)						A→G	het→hom
*rs341493267*	Loin fat with skin (kg)	1.64 ^a^	0.11	1.74 ^a^	0.05	1.92 ^b^	0.03	0.0477	**	ns	ns	+0.14 ^T^	-
	Daily gain (30–100 kg) (g)	998 ^A^	70	900 ^AB^	13	865 ^B^	11	0.0067	-	ns	ns	−67 **	-
	Daily gain (0–100 kg) (g)	698 ^a^	32	655 ^ab^	7	633 ^b^	7	0.0351	-	ns	ns	−33 *	-
	Feed conversion (kg/kg)	2.70 ^a^	0.14	2.90 ^ab^	0.034	3.02 ^b^	0.031	0.0165	-	ns	ns	+0.16 **	-
	Slaughter age (days)	173 ^a^	9	186 ^ab^	2	194 ^b^	2	0.0303	-	ns	ns	−6.72 ^T^	-
	Days in test (days)	92 ^a^	7	103 ^ab^	2	107 ^b^	1	0.0196	-	ns	ns	−4.37 ^T^	-
*rs346079334*		GG (*n* = 58)		GT (*n* = 56)		TT (*n* = 55)							
	Peritoneal fat (kg)	0.64 ^a^	0.02	0.60 ^b^	0.02	0.60 ^b^	0.03	0.0246	**	ns	ns	-	-
	Loin fat with skin (kg)	1.85 ^ab^	0.06	1.79 ^a^	0.06	1.84 ^b^	0.07	0.0356	***	ns	ns	-	-
	Loin eye height (cm)	7.29 ^ab^	0.07	7.43 ^a^	0.08	7.18 ^b^	0.08	0.0093	***	**	ns		−0.10 *
	Loin eye area (cm^2^)	58.7 ^ab^	0.77	60.1 ^a^	0.81	58.3 ^b^	0.77	0.0138	**	**	ns		−1.16 *
	pH loin45	6.12 ^A^	0.03	6.28 ^B^	0.03	6.19 ^AB^	0.04	0.0011	-	ns	ns	+0.03 ^T^	−0.07 **
*rs330154033*		AA (*n* = 9)		AG (*n* = 76)		GG (*n* = 84)							
	Carcass yield (kg)	74.7 ^a^	0.33	75.8 ^b^	0.14	75.8 ^b^	0.13	0.0415	**	ns	ns	+0.55 *	−0.26 *
	Loin mass (kg)	7.55 ^a^	0.13	8.13 ^b^	0.070	8.19 ^b^	0.066	0.0162	**	ns	ns	-	-
	Ham fat with skin (kg)	1.55 ^a^	0.14	1.84 ^b^	0.043	1.91 ^b^	0.033	0.0455	**	ns	ns	0.18 **	−0.067 ^T^
	Loin eye height (cm)	6.70 ^a^	0.17	7.34 ^b^	0.07	7.33 ^b^	0.06	0.0143	***	ns	ns	0.31 **	−0.16 *
	Loin eye area (cm^2^)	51.8 ^A^	2.67	59.8 ^B^	0.64	59.6 ^B^	0.64	0.0015	**	ns	ns	3.93 ***	−2.07 **
	Daily gain (30–100 kg) (g)	1034 ^A^	56	883 ^B^	14	875 ^B^	11	0.0009	-	ns	ns	−79 ***	
	Daily gain (0–100 kg) (g)	709 ^a^	27	647 ^ab^	8	625 ^b^	16	0.0207	-	ns	ns	−36 **	13.03 ^T^
	Feed conversion (kg/kg)	2.55 ^A^	0.081	2.92 ^AB^	0.033	3.01 ^B^	0.032	0.0002	-	ns	ns	0.23 ***	−0.07 *
	Slaughter age (days)	168 ^a^	8	189 ^b^	3	192 ^b^	2	0.0158	-	ns	ns	-	-
	Days in test (days)	87 ^A^	5	105 ^B^	2	105 ^B^	1	0.0025	-	ns	ns	-	-
*rs81333153*		CC (*n* = 10)		GC (*n* = 69)		GG (*n* = 90)							
	Feed conversion (kg/kg)	3.16a	0.068	2.97ab	0.033	2.92b	0.034	0.0482	-	ns	ns	−0.12 *	-

The same superscripts described values in the same statistical group (A,B = *p* < 0.01; a,b = *p* < 0.05), *p*-value in GLM significant * *p* < 0.05, ** *p* < 0.01, *** *p* < 0.001, ns—not significant, X2P—covariate for weight of the right side of the carcass. T— trend, *p*-value 0.05–0.20. het—heterozygous, hom—homozygous. Loin pH45 – pH in loin measured 45 minutes after slaughter.

## Data Availability

The sequence data for RNA libraries have been submitted to public database the Gene Expression Omnibus with the accession number GSE160436.

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
