# Peer review of "The Effect of BSCL2 Gene on Fat Deposition Traits in Pigs"

_animals, 2023, doi:10.3390/ani13040641_

Round 1

Reviewer 1 Report

The manuscript first calls the variation using RNA sequencing methods from which mutations in BSCL2 were selected as candidates to measure their effects on phenotypes, and found some interesting results. The manuscript is well written. 

1. The significant test for Figure 2C should be described in the section “materials and methods”.

2. Figures are obscure, please make them clearer.

Author Response

Dear Reviewer,

Thank you very much for your accurate review. The manuscript was corrected based on your suggestions.

Q1.The significant test for Figure 2C should be described in the section “materials and methods”. - The information of BSCL2 isoform expression pattern analysis was added to the section of “materials and methods”

Q2. Figures are obscure, please make them clearer. We understand that our Figures contain numerous details. So we attached an extra pdf file to see all details on the Figures. Nevertheless, we improved the Figures and Figure captions to be more precise. I hope that now they are clearer.

Reviewer 2 Report

1- BSCL2 stands for what. Note it.

2- Line 139:

The quality and quantity of the cDNA libraries were 139 assessed by the Qubit Fluorometer (Invitrogen) and the TapeStation 2200 system (D1000 140 tape, Agilent). 

Rewrite it as blow: 

The quantity and quality of the cDNA libraries were  assessed by the Qubit Fluorometer (Invitrogen) and the TapeStation 2200 system (D1000 140 tape, Agilent), respectively.

Author Response

Dear Reviewer,

Thank you very much for your accurate review. The manuscript was corrected based on your suggestions.

Q1, BSCL2 stands for what. Note it. – BSCL2 encodes seipin, a transmembrane endoplasmic reticulum protein it is described in the abstract and introduction section

Q2 - Line 139: - The quality and quantity of the cDNA libraries were 139 assessed by the Qubit Fluorometer (Invitrogen) and the TapeStation 2200 system (D1000 140 tape, Agilent).

Rewrite it as blow:

The quantity and quality of the cDNA libraries were  assessed by the Qubit Fluorometer (Invitrogen) and the TapeStation 2200 system (D1000 140 tape, Agilent), respectively.

This part/paragraph was removed according to Reviewer 3 suggestions because it was described in our previous study, so we added a reference in this place.

Reviewer 3 Report

The manuscript has some novel results regarding the role of BSCL2 in fat traits in pigs. Firstly, the title is a complete overstatement, there is no evidence from the manuscript to support the current conclusion of key determination. It is just based on the association analysis, not the knockdown of the genes to derive a such conclusion. Secondly, the authors mixed the results from the previous studies with the current one, which is not acceptable. Please clarify from the introduction what has been done and what objectives of the current one. Some bioinformatic results are not necessary and should move to the supplementary files.

Line 27: Define the abbreviation for FD.

Some abbreviations such as FD, LD, or AT might be not necessary.

Line 93: Add the province and country for the place.

Line 111-117: The authors might extend the text to describe how protocol and backfat were measured

Line 146-156: Since the results were derived from previous results, it is not necessary to repeat them. The authors should begin the manuscript with the bioinformatics sections.

Line 186: write the ijkl in the under script

What did the authors mean by “Outdoor gene”

Remove lines 200-201

Remove 3.1 and 3.2 since it is not completely the results of the current study. The enrichment analyses are not necessary and not the real results derived from experimental analyses.

 Tables: Please put the SE in the normal form, not superscript

Author Response

Dear Reviewer,

Thank you very much for your accurate review. The manuscript was partly corrected based on your suggestions.

Q1. The manuscript has some novel results regarding the role of BSCL2 in fat traits in pigs. Firstly, the title is a complete overstatement, there is no evidence from the manuscript to support the current conclusion of key determination. It is just based on the association analysis, not the knockdown of the genes to derive a such conclusion. -  the title was changed for “The effect of BSCL2 gene  on fat deposition traits in pigs” to be more precise

Q2. Secondly, the authors mixed the results from the previous studies with the current one, which is not acceptable. Please clarify from the introduction what has been done and what objectives of the current one. - – According to your suggestion, the goal of the present study was moved to the end paragraph to be more clear about what was the aim and what was investigated, I hope that this version is more clear. The previous study was separate from the aim of this study.

Q3. Some bioinformatic results are not necessary and should move to the supplementary files. – Presented bioinformatic results are not too extensively described according to our knowledge and are necessary to confirm reliable results obtained. Thus, we prefer not to move such results to the supplement. However, we can modify the Results section if you think it's necessary.

Q4. Line 27: Define the abbreviation for FD. – it was defined.

Q5. Some abbreviations such as FD, LD, or AT might be not necessary. – AT (adipose tissue) and FD (fat deposition) abbreviations are numerous used in the text body, so I think they are necessary. However, LD – lipid droplet was moved. Thank you for your suggestion.

Q6. Line 93: Add the province and country for the place. – this information was added.

Q7. Line 111-117: The authors might extend the text to describe how protocol and backfat were measured. This description was extended, moreover we added the reference where this protocol is broader described.

Q8. Line 146-156: Since the results were derived from previous results, it is not necessary to repeat them. The authors should begin the manuscript with the bioinformatics sections. It's right the RNA sequencing for fat tissue was described previously, so this part was significantly shortened and combined with the subsection considering GATK analysis (2.2). In this study, GATK analysis included fat RNA-seq data, it is a new analysis because our previous study presented GATK analysis included liver RNA-seq data, however content was similar, so this GATK analysis subsection was also significantly shortened. Thank you very much for your suggestion.

Q9. Line 186: write the ijkl in the under script – it was corrected

Q10. What did the authors mean by “Outdoor gene” – the wording was wrong, it was exchanged for off-gene variants

Q11. Remove lines 200-201 – this sentence was removed

Q12. Remove 3.1 and 3.2 since it is not completely the results of the current study. The enrichment analyses are not necessary and not the real results derived from experimental analyses. 3.1.Variant calling based on RNA-seq data there are results derived from current experimental analyses. It was not described before, just RNA-seq analysis, which was previously described in our study, where the detailed raw data processing was described, so here this information was omitted to avoid repetition, but variant calling analysis based on backfat RNA-seq data is fresh and firstly described. You may have had the wrong impression because our previous considering FGL1 study was based on RNA sequencing but on liver tissue. 3.2. Genes with missense variants functional analysis. Here are also presented new results concerning functional analysis for genes with missense variants expressed in fat tissue.

Round 2

Reviewer 3 Report

The authors have addressed my comments. However, the quality of the writing is not suitable for publication at this stage. Many sentences are difficult to understand; the authors should spend time proofreading. 

 “mutation effects within the candidate gene  BSCL2, in terms of their slaughter and fattening characteristics, and meat quality traits”

Might rewrite “  This study evaluated the effects of BSCL2 variants on slaughter and fattening characteristics, and meat quality traits”

“We suggested that proposed variant calling analysis using  RNA-seq data is useful to predict genetic markers in animals” It is still not clear, did the authors mean “ We suggested that proposed variant calling derived from

RNA-seq data might be useful genetics markers for complex traits in animals”

Line 31: Which studies? This study?

BSCL2 variants: Which one

Line 32: Which mutations? All of them

The authors might spend time checking the flow of the abstract, it is not smooth enough now.

Line 145: Add the reference for the tools

Line 248: It is unnecessary to repeat “identified using variant calling analysis “

Line 254: Are the 0.00055 chicq values or p values?

Line 257: it was shown that fully linked rs341493267 and  rs346079334 BSCL2 variants

Line 273 and others” Change the arrows to letters

In previous studies,  Studies in Duroc pigs: The authors should check plural forms of a subject such as studies or study (if only one reference is used), etc.

it a major functional candidate [38]: it is not exact

Author Response

Dear reviewer,

The manuscript, before submission, was corrected by the English Editing service MDPI. However, we reviewed our manuscript once more, and all sentences which were not precisely clear were reorganized. Therefore I hope that the manuscript is much clearer in this version. All reviewer suggestions were included.

 The authors have addressed my comments. However, the quality of the writing is not suitable for publication at this stage. Many sentences are difficult to understand; the authors should spend time proofreading.

 “mutation effects within the candidate gene  BSCL2, in terms of their slaughter and fattening characteristics, and meat quality traits”, Might rewrite “  This study evaluated the effects of BSCL2 variants on slaughter and fattening characteristics, and meat quality traits” – the sentence was changed.

“We suggested that proposed variant calling analysis using  RNA-seq data is useful to predict genetic markers in animals” It is still not clear, did the authors mean “ We suggested that proposed variant calling derived from – the sentence was rearranged.

RNA-seq data might be useful genetics markers for complex traits in animals” – the suggestion was included

Line 31: Which studies? This study? – this information was added

BSCL2 variants: Which one- this information was added

Line 32: Which mutations? All of them - this information was added

The authors might spend time checking the flow of the abstract, it is not smooth enough now. – the abstract was thoroughly corrected.

Line 145: Add the reference for the tools – the references were added

Line 248: It is unnecessary to repeat “identified using variant calling analysis “ – it was removed

Line 254: Are the 0.00055 chicq values or p values? – this was corrected

Line 257: it was shown that fully linked rs341493267 and  rs346079334 BSCL2 variants – the sentence was chenged

Line 273 and others” Change the arrows to letters – the arrows were changed for letters

In previous studies,  Studies in Duroc pigs: The authors should check plural forms of a subject such as studies or study (if only one reference is used), etc. – it was corrected

it a major functional candidate [38]: it is not exact – this sentence was chenged

Round 3

Reviewer 3 Report

Dear Authors, 

Thank you for responding to my comments. The manuscript has been improved. 

Author Response

Dear Reviewer,

Thank you for spending your time on the correction of my manuscript.